# New Emerging Targets in Cancer Immunotherapy: The Role of B7-H3

**DOI:** 10.3390/vaccines12010054

**Published:** 2024-01-05

**Authors:** Ioannis-Alexios Koumprentziotis, Charalampos Theocharopoulos, Dimitra Foteinou, Erasmia Angeli, Amalia Anastasopoulou, Helen Gogas, Dimitrios C. Ziogas

**Affiliations:** First Department of Medicine, Laiko General Hospital, School of Medicine, National and Kapodistrian University of Athens, 11527 Athens, Greece; giannhskmpr@gmail.com (I.-A.K.); hartheoch@gmail.com (C.T.); foteinoudim@gmail.com (D.F.); erasmia.angeli@gmail.com (E.A.); amanastasop@yahoo.gr (A.A.); helgogas@gmail.com (H.G.)

**Keywords:** B7-H3, immune checkpoints, checkpoint inhibitors, immunotherapy, ADCs, solid malignancies

## Abstract

Immune checkpoints (ICs) are molecules implicated in the fine-tuning of immune response via co-inhibitory or co-stimulatory signals, and serve to secure minimized host damage. Targeting ICs with various therapeutic modalities, including checkpoint inhibitors/monoclonal antibodies (mAbs), antibody-drug conjugates (ADCs), and CAR-T cells has produced remarkable results, especially in immunogenic tumors, setting a paradigm shift in cancer therapeutics through the incorporation of these IC-targeted treatments. However, the large proportion of subjects who experience primary or secondary resistance to available IC-targeted options necessitates further advancements that render immunotherapy beneficial for a larger patient pool with longer duration of response. B7-H3 (B7 Homolog 3 Protein, CD276) is a member of the B7 family of IC proteins that exerts pleiotropic immunomodulatory effects both in physiologic and pathologic contexts. Mounting evidence has demonstrated an aberrant expression of B7-H3 in various solid malignancies, including tumors less sensitive to current immunotherapeutic options, and has associated its expression with advanced disease, worse patient survival and impaired response to IC-based regimens. Anti-B7-H3 agents, including novel mAbs, bispecific antibodies, ADCs, CAR-T cells, and radioimmunotherapy agents, have exhibited encouraging antitumor activity in preclinical models and have recently entered clinical testing for several cancer types. In the present review, we concisely present the functional implications of B7-H3 and discuss the latest evidence regarding its prognostic significance and therapeutic potential in solid malignancies, with emphasis on anti-B7-H3 modalities that are currently evaluated in clinical trial settings. Better understanding of B7-H3 intricate interactions in the tumor microenvironment will expand the oncological utility of anti-B7-H3 agents and further shape their role in cancer therapeutics.

## 1. Introduction

Cancer cells harbor multiple mutations that are phenotypically manifested by the expression of tumor-associated neoantigens which can subsequently prime a neoantigen-specific anti-tumor immune response [1]. Effective tumor immunosurveillance requires adequate neoantigen expression and efficient sequential interactions between different immune cell subsets and tumor microenvironment (TME) constituents. However, tumor cells leverage several intricate phenomena of the immune system to eventually render themselves non-responsive to the immune attack. Immune checkpoints (ICs) are molecules implicated in the fine-tuning of immune responses through co-inhibitory or co-stimulatory signals that serve to restrict host damage [2]. Chronic neoantigen exposure and sustained T-cell stimulation results in IC upregulation and eventual T-cell exhaustion. These alterations in the immune-mediated equilibrium favor evasion from cancer cell destruction and eventually drive disease progression. Currently, used immunotherapeutic agents targeting ICs aim to halt co-inhibitory T-cell signaling and reinstate a functional antineoplastic immune response. The therapeutic utility of IC manipulation has been showcased by the first-generation checkpoint inhibitors (CPIs), including anti-PD-1/PD-L1 and anti-CTLA-4 mAbs. These agents have been incorporated in treatment algorithms for many solid malignancies achieving durable responses either as monotherapies or in combinatorial regimens. However, a considerable percentage of subjects experience minimal benefit or present with early relapse owing to numerous events that curtail CPI-enhanced immunosurveillance. Upregulation of additional IC molecules beyond PD-1/PD-L1 and CTLA-4 was a well-described mechanism of resistance that has fueled the research interest on identifying novel ICs that could improve treatment outcomes and render immunotherapy beneficial for a larger set of cancer patients [3,4,5]. In the present review, we focus on the role of B7-H3 in different solid malignancies. B7-H3 (B7 Homolog 3 Protein, CD276) is a multifunctional IC molecule that exerts both co-stimulatory and co-inhibitory effects. Mounting data suggest a pleiotropic activity through extensive immune regulation, enhanced tumor cell proliferation, invasiveness, and metastatic potential. Importantly, contrary to non-cancerous tissue, B7-H3 is significantly upregulated in a plethora of solid tumors [6]. Figure 1 depicts data from the Human Protein Atlas regarding B7-H3 mRNA expression in an array of malignant diseases (Figure 1) [7]. Following the immunologic and non-immunologic properties of B7-H3 (Table 1), we discuss here B7-H3 expression patterns (Appendix A), its impact in the evolution of various cancer types, its association with some major clinicopathologic characteristics as well as its prognostic significance and therapeutic potential in these solid malignancies. To this end, we summarize all latest evidence from preclinical data to clinical trial results on the B7-H3 targeted modalities (e.g., novel mAbs, bispecific antibodies, ADCs, CAR-T cells, and radioimmunotherapy agents) (Figure 2) that are currently investigated in anticancer research (Table 2).

### 1.1. B7-H3 Structure and Receptors

B7-H3 comprises a 316-amino acids long type I transmembrane protein with two isoforms: 2Ig B7-H3 and 4Ig B7-H3. B7-H3 belongs to the B7 family of proteins and shares a 20–27% identical amino acid sequence with the rest of the B7 family members [8]. The human B7-H3 gene is found on chromosome 15q24.1, whereas its murine ortholog is located in chromosome 9. The 4Ig B7-H3 is the main isoform in humans and consists of two pairs of extracellular IgV-like and IgC-like domains, a transmembrane region, and a 45-amino acids long cytoplasmic tail [9]. Although B7-H’s receptors are yet to be conclusively documented, currently identified receptors include: (i) triggering receptors expressed in myeloid cells (TREM)-like transcript 2 (TLT-2); (ii) IL-20 receptor subunit α (IL-20Rα), and (iii) phospholipase A2 receptor 1 (PLA2R1) [10]. TLT-2 expression has been reported in neutrophils, macrophages, B-cells, NK cells, DCs, CD8^+^, and CD4^+^ T-cells [11,12]. Contradictory data have been published on the functional significance of B7-H3/TLT-2 engagement, as Hashiguchi et al. [12], and Kobori et al. [13] described heightened proliferation and effector function of CD8^+^ T-cells, whereas later studies opposed B7-H3/TLT-2 interaction. Leitner et al., in particular, reported no evidence of TLT-2 and B7-H3 interaction and concluded B7-H3 mediated inhibition of CD4^+^ and CD8^+^ T-cell proliferation and markedly reduced cytokine production [14]. Negative results on a potential TLT-2/B7-H3 binding were also presented by Yan et al. [15] and Vigdorovich et al. [16] solidifying the need for further exploration of the presumed engagement TLT-2 by B7-H3. Regarding IL-20Rα, although its interaction with B7-H3 has been proposed in different studies [17,18], the lack of expression in immune cells [19], necessitated additional research to elucidate the B7-H3/IL-20Rα immunomodulatory potential. Expression of IL-20Rα has been reported in several healthy human tissues including skin, prostate, testis, and heart. Importantly, recent data indicate a role for IL-20Rα in the formation of an immunosuppressive TME through JAK1-STAT3-SOX2-mediated upregulation of PD-L1, expansion of MDSCs, and subsequent restriction of TILs and NK cells [20]. PLA2R1 was identified by Cao et al. as another possible B7-H3 receptor; however, the significance of this interaction remains to be unraveled. PLA2R1 downregulation has been described in several malignancies, suggesting it might function as a tumor suppressor [21]. To this end, PLA2R1 expression was reported to suppress tumorigenesis through PARP1 restriction [22] and activate p53 to promote cellular senescence [23].

### 1.2. B7-H3 Localization

B7-H3 presence has been confirmed both intracellularly, localized in the nucleus, cytoplasm, or cell membrane, as well as extracellularly as soluble B7-H3 (sB7-H3). Importantly, specific patterns of expression have been described depending on the given neoplasm, while evidence of prognostic significance based on the cellular localization has also emerged. In healthy tissues, immunohistochemical analysis has revealed weak staining and primarily cytoplasmic localization of B7-H3 [24]. In the malignant setting, B7-H3 is mainly localized in the cytoplasm and on the cell membrane and to a lesser degree within the nucleus. In colorectal cancer, B7-H3 was primarily found in the cytoplasm (62%) or cell membrane (46%), while nuclear expression was noted in 30% [25,26], and stromal presence was reported in 77% [26]. Importantly, in patients with nuclear expression 5-year OS was significantly reduced compared with nuclear negative subjects (62% vs. 79%, respectively, *p* = 0.004) [25]. In clear cell renal cell carcinoma, membranous expression was the most prevalent and conferred improved survival outcomes compared to cytoplasmic localization [27]. In pancreatic adenocarcinoma cells, B7-H3 was mainly observed in the cytoplasm whereas in stroma cells, B7-H3 was primarily present on the cell membrane [28]. Notably, cervical and gastric carcinomas exhibit greater B7-H3 expression in stromal compartments compared to tumor cells [29,30]. This is of particular importance in the development of stromal targeting therapies, such as non-internalizing, stroma-targeting ADCs [31], that are able to circumvent tumor heterogeneity and exert their cytotoxicity through local diffusion. The soluble form of B7-H3 is produced either through alternative splicing [32] or though cleavage by matrix metalloproteases from the surface of B7-H3^+^ cells [33]. In the study of Zhang et al., sB7-H3 was detected in the totality of the studied normal human serum samples and was also determined to be functionally active [33]. In the context of malignancy, sB7-H3 has emerged as a prognostic biomarker, as discussed in the following sections. Importantly, the levels of sB7-H3 could also influence the efficacy of anti-B7-H3 agents. Minimal shedding is a desired feature of target antigens for ADCs, as increased levels of circulating antigens bind ADCs and render them ineffective. This on-target off-tumor interaction restricts the proportion of the administered immunoconjugate that reaches the TME and curtails the overall efficiency [34]. Similarly, sB7-H3 could limit the effectiveness of anti-B7-H3 mAbs, acting as decoy and resulting in on-target off-tumor binding. To this end, several studies have reported worse outcomes upon anti-PD-1 or anti-PD-L1 treatment in subjects with high sPD-L1 or sPD-1 [35,36,37].

### 1.3. B7-H3 Immunologic Functions

B7-H3 appears to exert both costimulatory and coinhibitory effects on T-cells (Figure 3). In the study that first presented B7-H3, Chapoval et al. [8] reported upon anti-CD3 stimulation, addition of a B7-H3-Ig fusion protein increased proliferation of both CD4^+^ and CD8^+^T-cells and selectively augmented IFN-γ production. A putative B7-H3 counterreceptor was shown to be expressed in activated T-cells [8]. However, its exact identity is still to be unraveled. Similar results were subsequently published by Hashiguchi et al. and Kobori et al., who proposed TLT-2 as a B7-H3 counterreceptor [12,13]. Synchronously, Suh et al. and Prasad et al. [38] provided evidence of the coinhibitory activity of B7-H3, reporting decreased T-cell activation and proliferation. In the latter study, these effects were attributed to reduced IL-2 production, presumably via B7-H3-induced diminished transcriptional activity of NFAT, NF-kB, and AP-1 [38]. Interestingly, addition of exogenous IL-2 fully reversed these findings [38]. Leitner et al., subsequently, reproduced these results, while observing no evidence supporting B7-H3/TLT-2 interaction [14]. Furthermore, co-culture of CD3^+^T-cells with B7-H3^+^ breast cancer cells resulted in IFN-γ downregulation through decreased PI3K/AKT/mTOR signaling [39]. Tumor residing B7-H3^+^ DCs, derived from lung adenocarcinoma patient samples, exhibited significantly restricted T-cell activation capacity, decreased IL-12, and increased IL-10 secretion [40]. B7-H3 was also shown to suppress NK cell activity, as human NK cells co-cultured with B7-H3^+^ neuroblastoma cells displayed significantly restricted cytotoxicity [41]. In macrophages, B7-H3 promotes acquisition of the immunosuppressive M2 phenotype via activation of the JAK2/STAT3 pathway [42,43]. sB7-H3 was shown to enhance chemotaxis of MDSCs, contributing to the formation of an immunosuppressive tumor milieu [44]. Furthermore, B7-H3 expression in renal cell carcinoma cells and vasculature was positively associated with the number of tumor infiltrating Tregs [45].

### 1.4. B7-H3 Non-Immunologic Functions

Tumor cell expression of B7-H3 confers multifunctional effects that drive disease progression and render expressing cells resistant to treatment. B7-H3 upregulation was shown to expand breast cancer stem cells through induction of the MAPK/ERK signaling pathway, while also disrupting cell polarity and promoting invasiveness [46]. B7-H3 expression in colorectal cancer cell lines, promoted cell proliferation via upregulation of Cyclin D1 and CDK4 [47], while silencing of B7-H3 resulted in markedly reduced tumor cell multiplication in lung adenocarcinoma cells [48]. B7-H3 has also been shown to induce epithelial-to-mesenchymal transition through the JAK2/STAT3/SLUG [49] and PI3K/AKT pathways [50]. Enhanced invasiveness is further granted via B7-H3-induced MMP-2/-9 upregulation [51,52]. B7-H3 expression has also been linked to apoptosis evasion, through downregulation of pro-apoptotic proteins including p53, Bac, Caspase 8, Rb, and upregulation of anti-apoptotic molecules including E7, p16, p21, BCL-2, and BCL-XL [53,54]. B7-H3 is also overexpressed on tumor-associated endothelial cells and induces abnormal angiogenesis via overexpression of proangiogenic molecules including IL-1, IL-6, and vascular endothelial growth factor (VEGF) [55,56]. Furthermore, B7-H3 reprograms the metabolism of glucose and lipids and modulates the metabolic influx of tumor cells to promote tumorigenesis. Particularly, it promotes glucose intake through upregulation of HIF-1a via PI3K/AKT pathway and induces a hyperglycolytic state mainly through hexokinase 2 induction [57]. Lastly, B7-H3 expression has been associated with resistance to radiotherapy and upon administration of chemotherapy drugs including gemcitabine, paclitaxel, doxorubicin, oxaliplatin, and 5FU [58,59,60,61,62,63,64].

**Table 1 vaccines-12-00054-t001:** Summary of main immunologic and non-immunologic functions of B7-H3.

Type of Function	Function	Mechanism	Tumor	Reference
Immunologic	Suppression of T-cell proliferation	Decreased PI3K/AKT/mTOR signaling	Breast cancer	[39]
	Suppression of T-cell activation	Restriction of DC antigen-presenting and T-cell activation capacity	Lung cancer	[40]
	Suppression of T-cell cytotoxicity	Decreased PI3K/AKT/mTOR signaling	Breast cancer	[39]
	Suppression of NK cell cytotoxicity	Downregulation of IFN-γ	Neuroblastoma	[41]
	Modulation of TAM differentiation	Induction of the immunosuppressive M2 polarization through STAT3 signaling	Liver cancer	[43]
Non-immunologic	Promotion of cancer cell invasion and migration	Upregulation of MMP2 and MMP9	Liver cancer	[49]
			Colorectal cancer	[65]
		Promotion of EMT via the JAK2/STAT3/SLUG signaling pathway	Liver cancer	[49]
		Upregulation of IL-8 through the TLR4/NF-Kb pathway	Pancreatic cancer	[66]
	Enhancement of cancer cell proliferation	Upregulation of Cyclin D1 and CDK4	Colorectal cancer	[47]
	Expansion of cancer stem cell population	Induction of the MAPK/ERK signaling axis	Breast cancer	[46]
	Enhancement of cancer cell metabolism	Promotion of aerobic glycolysis through upregulation of hexokinase 2	Colorectal cancer	[67]
		Enhancement of glucose uptake and lipid metabolism	Lung cancer	[68]
	Apoptosis evasion	Downregulation of P53 and Caspase 3 and BAX apoptotic proteins through the PI3K/AKT pathway	Gastric cancer	[54]
		Regulation of the apoptosis-related proteins P21, E7, Rb, P16, PARP-1, Caspase 8, BAX, BCL-2, BCL-XL	Cervical cancer	[53]
	Stimulation of tumor angiogenesis	Upregulation of VEGF through the TLR4/NF-Kb pathway	Pancreatic cancer	[66]
	Induction of treatment resistance	Gemcitabine resistance through upregulation of survivin	Pancreatic cancer	[58]
		Radiotherapy resistance through induction of PI3K/AKT signaling	Gastric cancer	[60]
		Radiotherapy resistance through regulation of KIF15-activated ERK1/2	Colorectal cancer	[59]
		Oxaliplatin resistance via regulation of CDC25A expression through the STAT3 signaling	Colorectal cancer	[64]
		5-FU resistance via regulation of CDC25A expression through the STAT3 signaling	Colorectal cancer	[64]
		Doxorubicin resistance through regulation of the AKT/TM4SF1/SIRT1 pathway.	Colorectal cancer	[63]
		Paclitaxel resistance through upregulation of the PI3K/AKT/BCL-2 signaling pathway	Ovarian cancer	[69]
		Paclitaxel resistance through upregulation of the JAK2/STAT3 pathway	Breast cancer	[61]

**Table 2 vaccines-12-00054-t002:** Summary of completed, active, and recruiting clinical trials evaluating B7-H3-targeting agents in solid malignancies.

ClinicalTrials.gov Identifier	Agent	Format	Status	Phase	Indications
NCT02381314	Enoblituzumab (MGA271) + Ipilimumab	Humanized IgG1 Anti-B7-H3 monoclonal antibody+ Humanized IgG1 Anti-CTLA-4 antibody	Completed	I	Melanoma, NSCLC, and other B7-H3^+^ solid tumors
NCT02475213	Enoblituzumab (MGA271) + Pembrolizumab Or Enoblituzumab (MGA271) + retifanlimab (MGA012)	Humanized IgG1 Anti-B7-H3 monoclonal antibody + Anti-PD-1 monoclonal antibody Or Humanized IgG1 Anti-B7-H3 monoclonal antibody + Anti-PD-1 monoclonal antibody	Completed	I	B7-H3^+^ melanoma, squamous cell carcinoma of the head and neck, NSCLC, urothelial cancer and other B7-H3^+^ refractory cancers
NCT01391143	Enoblituzumab (MGA271)	Humanized IgG1 Anti-B7-H3 monoclonal antibody	Completed	I	Prostate cancer, melanoma, RCC, TNBC, head and neck cancer, bladder cancer, NSCLC
NCT02982941	Enoblituzumab (MGA271)	Humanized IgG1 Anti-B7-H3 monoclonal antibody	Completed	I	B7-H3^+^ solid tumors in children and young adults
NCT03406949	Orlotamab (MGD009) + retifanlimab (MGA012)	Humanized B7-H3 CD3 Dual-Affinity Re-Targeting (DART) Protein + Anti-PD-1 monoclonal antibody	Completed	I	B7-H3^+^ tumors
NCT00089245	^131^I-omburtamab	B7-H3-Targeted Radiolabeled Monoclonal Antibody Therapy	Active, not recruiting	I	Sarcoma, neuroblastoma, brain and CNS tumors
NCT05276609	HS-20093	Humanized IgG1 Anti-B7-H3 ADC	Recruiting	I	Advanced solid tumors
NCT05293496	Vobramitamab duocarmazine (MGC018) + Lorigerlimab (MGD019)	Humanized IgG1 Anti-B7-H3 ADC + bispecific DART^®^ molecule that binds PD-1 and CTLA-4	Recruiting	I	Advanced solid Tumors
NCT05190185	TAA06	B7-H3-Targeted CAR-T Cells	Recruiting	I	Advanced solid tumors including melanoma, lung cancer, or colorectal cancer
NCT04897321	B7-H3-CAR-T cells + Fludarabine + Cyclophosphamide + Mesna	B7-H3-Targeted CAR-T Cells + Lymphodepletion chemotherapy	Recruiting	I	B7-H3^+^ solid tumors
NCT04483778	Second generation 4-1BBζ B7H3-EGFRt-DHFR	B7-H3-Targeted CAR-T Cells (Arm A CAR-T-cells include the protein EGFRt and Arm B CAR-T-cells include the protein HER2tG)	Recruiting	I	Recurrent/refractory solid tumors in children and young adults
NCT04842812	TILs/CAR-TILs	TILs/CAR-TILs targeting B7-H3 and other molecules and with knockdown of PD-1	Recruiting	I	Advanced solid tumors
NCT03198052	GPC3/Mesothelin/Claudin18.2/GUCY2C/B7-H3/PSCA/PSMA/MUC1/TGFβ/HER2/Lewis-Y/AXL/EGFR CAR-T cells	GPC3/Mesothelin/Claudin18.2/GUCY2C/B7-H3/PSCA/PSMA/MUC1/TGFβ/HER2/Lewis-Y/AXL/EGFR-targeted CAR-T cells	Recruiting	I	Lung cancer and other cancers
NCT05341492	EGFR/B7-H3 CAR-T cells	EGFR/B7-H3-targeted CAR-T cells	Recruiting	I	EGFR/B7-H3^+^ Lung cancer and TNBC
NCT04145622	Ifinatamab Deruxtecan (I-DXd, DS-7300)	Humanized IgG1 Anti-B7-H3 monoclonal ADC	Recruiting	I/II	Advanced solid tumors
NCT05323201	fhB7H3.CAR-T cells + Fludarabine + Cyclophosphamide	B7-H3-Targeted CAR-T Cells + Lymphodepletion chemotherapy	Recruiting	I/II	Hepatocellular carcinoma
NCT05143151	B7-H3 CAR-T-cells	B7-H3-Targeted CAR-T Cells	Recruiting	I/II	Advanced pancreatic carcinoma
NCT04637503	GD2, PSMA and B7-H3 CAR-T cells	GD2, PSMA and B7-H3-Targeted CAR-T cells	Recruiting	I/II	Relapsed and refractory neuroblastoma
NCT04432649	4SCAR-276	4th generation lentiviral B7-H3-Targeted CAR-T cells	Recruiting	I/II	B7-H3^+^ solid tumors
NCT02923180	Enoblituzumab (MGA271)	Humanized IgG1 Anti-B7-H3 monoclonal antibody	Active, not recruiting	II	Localized intermediate and high-risk prostate cancer
NCT05551117	Vobramitamab duocarmazine (MGC018)	Humanized IgG1 Anti-B7-H3 ADC	Recruiting	II	Metastatic castration resistant prostate cancer
NCT05280470	Ifinatamab Deruxtecan (I-DXd, DS-7300)	Humanized IgG1 Anti-B7-H3 monoclonal ADC	Active, not recruiting	II	Extensive-stage SCLC
NCT03275402	131I-omburtamab	B7-H3-Targeted Radiolabeled Monoclonal Antibody Therapy	Active, not recruiting	II/III	Neuroblastoma and central nervous system/leptomeningeal metastases

Abbreviations: NSCLC: non-small cell lung carcinoma; SCLC: small cell lung carcinoma; CNS: central nervous system RCC: renal cell carcinoma; TNBC: triple negative breast cancer; CAR: chimeric antigen receptor; TILs: Tumor-Infiltrating Lymphocytes; ADC: antibody-drug conjugate.

## 2. Hepatopancreatic and Gastrointestinal Malignancies

### 2.1. Hepatocellular Carcinoma (HCC)

The functional implications and the therapeutic utility of B7-H3 in HCC has been thoroughly investigated by a plethora of recent studies, pinpointing its promising role as a prognostic biomarker and therapeutic target. Although B7-H3 protein expression is suppressed in the non-malignant setting, via extensive miRNA-mediated post-transcriptional mRNA modifications, healthy human liver tissue displayed the highest percentage of B7-H3 positivity (67%) among an array of other normal tissues [6]. In the same study, overexpression of B7-H3 was detected in 80% of HCC samples. Higher positivity was reported in the studies of Sun et al. (93.8%) [70], Wang et al. (88.57%) [71], and Liu et al. (90.5%) [72]. In serum, sB7-H3 has been proposed as a valuable biomarker for the identification of cirrhotic patients in high risk of developing HCC as well as for the early diagnosis of an arising HCC in a cirrhotic background (sensitivity: 76.5% and specificity: 93.1% with a cut-off value of 48.34 ng/mL) [73]. Interestingly, sB7-H3 has emerged as a better diagnostic marker than a-fetoprotein, CA19-9, and CA125 [73]. High expression of B7-H3 on HepG2 cells significantly promotes cell proliferation, adhesion, migration, and invasion capacity, while it inhibits the proliferation of CD8^+^T-cells [71] and decreases the number of CD8^+^ TILs [72,74]. These B7-H3 pro-metastatic effects primarily stem from upregulated expression and activity of MMP2/MMP9, as well as from promotion of EMT transition via the JAK2/STAT3/SLUG signaling [49]. Shrestha et al. showed siRNA-mediated knockdown of B7-H3 reversed TGF-*β*1 driven EMT and significantly restricted HCC metastatic potential [75]. According to the Cancer Genome Atlas—Liver Hepatocellular Carcinoma (TCGA-LIHC) database, HCC prognosis was significantly worse in subjects with the CD8^+^T-cell-low/B7-H3-high immunophenotype compared to other immunophenotypes [76]. Induction of the immunosuppressive M2 polarization of tumor-associated macrophages, via the STAT3 signaling pathway, further adds to the B7-H3-mediated negative regulation of T-cell antitumor response [43]. Zhou et al. demonstrated when HCC cells pretreated with anti-B7-H3 mAbs were co-cultured with activated T-cells, the latter exhibited significantly heightened killing efficacy and IFNγ and TNFα secretion, compared to T-cells in the treatment naïve group [74]. In the same study, B7-H3 neutralization in HCC-bearing mouse models, significantly reduced tumor growth and prolonged survival. In human HCC biopsies, B7-H3 positive immunostaining was significantly associated with advanced clinicopathological features and invasive behavior such as advanced stage, vascular infiltration, and presence of satellite lesions, indicative of intrahepatic metastasis [49,73,77]. In agreement, B7-H3 expression was more prevalent among patients with de novo metastatic HCC compared to subjects with localized disease (96% vs. 76.3%, *p* = 0.036) [78] and B7-H3^+^ patients had significantly restricted median overall survival (mOS) following surgery compared to B7-H3^−^ counterparts (19.2 vs. 37.3 months, respectively; *p* = 0.009) [78]. The negative prognostic impact of B7-H3 was also described by Kang et al. (mOS: 37.1 vs. 29.2, *p* = 0.011); and importantly, its poor prognosis was reproduced even after adjustment for other negative factors (*p* = 0.009) [49]. Increased sB7-H3 levels have also been associated with worse post-surgical outcomes (*p* = 0.048) [73]. All these abovementioned findings further support the rationale behind the focus on novel strategies targeting the B7-H3 pathway.

However, limited published data exist on the therapeutic implications of modalities targeting B7-H3 in patients with HCC. Vobramitamab duocarmazine (MGC018), is an ADC consisting of a humanized anti-B7-H3 mAb tethered to a duocarmycin analogue [79] and is currently being tested in a phase I trial (NCT05293496) in combination with lorigerlimab (MGD019), a bispecific dual affinity re-targeting (DART) molecule [80] directed against PD-1 and CTLA-4, in relapsed or refractory solid tumors, including HCC. Lastly, CAR T-cell immunotherapy targeting B7-H3 in HCC is under investigation by a phase I/II trial, in combination with fludarabine and cyclophosphamide (NCT05323201), and for hepatoblastoma in two phase I trials (NCT04897321, NCT04483778).

### 2.2. Pancreatic Adenocarcinoma (PAC)

In PAC patient specimens, the reported B7-H3 positivity rate ranges from 66% [28,81] to 88% [82] and 93.2% [83]. Significantly higher expression was identified in poorly differentiated compared to moderately and well differentiated disease (96.2% vs. 50%, *p* = 0.002 [52], 64.5% vs. 40%, *p* = 0.02 [84], 53.8% vs. 28.8% [85]). Regarding lymph node involvement and pathological stage, both statistically significant [52,83] and non-significant [84,85] correlations have been reported. Zhao et al. observed significantly higher invasion and migration capacity in B7-H3-high human PAC cell lines compared to B7-H3-low counterparts, while Xie et al. clarified B7-H3 promoted invasion and metastasis through the TLR4/NF-κB pathway in PAC cells [66]. These findings were subsequently replicated in vivo by subcutaneous transplantation tumor and orthotopic transplantation of PAC mouse models [86].

Increased B7-H3 expression was associated with significantly lower 2- and 5-year OS% (57%/23% vs. 34%/12%, *p* = 0.0072) and DFS% (38%/23% vs. 15%/8%, *p* = 0.0005) after pancreatectomy. Importantly, in the B7-H3-high subgroup, the 2-year DFS% was 0% compared to 18% in the low and 15% in the intermediate subgroups (*p* = 0.001) [81]. In another study, significantly lower PFS was described in B7-H3^+^ patients (*p* = 0.003) [84]. B7-H3 was shown to reduce gemcitabine-induced cytotoxity in human PAC cells [87], through upregulation of the anti-apoptotic survivin protein [58], while B7-H3 silencing restored gemcitabine chemosensitivity [58].

Therapeutically, the administration of anti-B7-H3 mAb in murine PAC models resulted in profound CD8^+^T-cell infiltration compared to baseline (*p* < 0.001),and significant reduction in tumor volume. Co-administration of anti-B7-H3 mAb with gemcitabine resulted in greater anti-tumor effect compared to either monotherapy (*p* < 0.0001) [83]. Radioimmunotherapy with a ^212^Pb-labeled anti-B7-H3 mAb [88] and B7-H3-targeting CAR-T cells [89] have also shown potent anti-PAC activity. B7-H3-targeted CAR-T cells are currently being tested in patients with advanced PAC in a phase I/II clinical trial (NCT05143151). More recently, Lutz et al. described notable anti-tumor efficacy of a B7-H3Xcd3 bispecific antibody in PAC, as evidenced by enhanced T-cell activation and secretion of IL-2, IFN-g, and perforin, which resulted in tumor cell lysis [90]. Lastly, vobramitamab duocarmazine is currently being tested in combination with lorigerlimab for patients with PAC in a phase I study (NCT05293496).

### 2.3. Gastric Cancer (GC)

Gastric cancer comprises another malignancy with documented B7-H3 expression, widely ranging from 58.8% [91] to 76% [29] and 78% [92]. B7-H3 was primarily found on the tumor stroma and in a lesser degree on tumor cells (82% vs. 18%) [29], while B7-H3^+^ stromal compartments were also more likely to have high rather than low levels of B7-H3 expression compared to B7-H3^+^ tumor cells (62.7% vs. 18.7%) [93]. B7-H3 was associated with tumor stage (*p* = 0.002), lymph node involvement (*p* = 0.006), and infiltration depth (*p* = 0.001 [93], *p* = 0.005 [94]). In the latter study, shRNA-mediated B7-H3 silencing in the N87 gastric cancer cell line suppressed cell migration and invasion in vitro and in vivo; downregulated metastasis-associated CXCR4; and inhibited PI3K/AKT, MAPK/ERK, and JAK2/STAT3 phosphorylation [94]. In accordance, induction of PI3K/AKT signaling was noted upon interaction of B7-H3 with fibronectin, resulting in inhibition of apoptosis and induction of evasion of GC cells [54]. AKT downstream stimulates NRF2 that transcriptionally activates glutathione biosynthetic genes [95]. Glutathione accumulation mediates resistance to oxidative stress and facilitate anchorage-independent proliferation and invasive behavior [96]. In GC cells, Lu et al. demonstrated B7-H3 was a key regulator of glutathione metabolism via AKT/NRF2 pathway and its expression in patient GC tissues was significantly associated with NRF2 expression (*p* = 0.04). Importantly, patients with high NRF2 expression had a significantly worse prognosis compared to NRF2-low subjects (*p* = 0.05) [97]. This sequence was effectively reversed upon B7-H3 knockout, which further promoted the expression of p53 and Caspase 3 apoptotic proteins and inhibited tumor growth in vivo [54]. P53 has been identified as an essential downstream target of the B7 family in GC [98]. Furthermore, B7-H3^+^ lesions exhibited significantly lower CD8^+^T-cell intratumoral density compared to B7-H3^−^ lesions (*p* = 0.0162) [92], while high B7-H3 expression was correlated with lower median number of CD8^+^T-cells in the tumor center, compared to low expression (86.4/mm^2^ vs. 157.7/mm^2^) [29]. Interestingly, B7-H3 expression increases the radiotherapy resistance of GC cells through regulating baseline levels of cell autophagy [60].

In terms of prognosis, Zhan et al. demonstrated patients with increased B7-H3 expression had significantly worse prognosis (*p* = 0.012), while the OS of the B7-H3 high group was poorer compared to B7-H3 low group, for both tumor and stromal expression (*p* = 0.007 and *p* = 0.048, respectively) [93]. Co-expression of B7-H3 with HER2 was also correlated with poor prognosis (*p* = 0.007) [99]. Despite the theoretical basis, few approaches targeting of B7-H3 have been described to date. Lutz et al. presented a bispecific antibody co-targeting CD3 and B7-H3 (CC-3) that produced promising antitumoral activity. Significant increase in CD4^+^ and CD8^+^T-cell proliferation was observed upon treatment with CC-3 in co-cultures of GC cells and T-cells (*p* = 0.01 and *p* = 0.001, respectively). Furthermore, significant increase was shown in secretion of IL-2, IFNγ, IL-10, and TNF that resulted in enhanced GC cell lysis (*p* < 0.0001) [90]. Combined inhibition of HER2 and B7-H3 using the humanized anti-HER2 mAb trastuzumab and an anti-B7-H3 mAb yielded better tumor control than either monotherapy [99]. B7-H3-targeting CAR-T cells also showed robust cytotoxic activity against GC cell lines both in vitro and in vivo [100].

### 2.4. Colorectal Cancer (CRC)

Many studies have supported the association between B7-H3 expression and chemotherapy resistance/progression in CRC. Its positivity rate was estimated at 77.95% in a large meta-analysis of 1200 CRC cases [101] and 50.8% in a more recent study of 805 CRC specimens [102]. B7-H3 expression was consistently more pronounced in cancerous tissue compared to adjacent healthy colon [103,104] and sB7-H3 was also significantly higher in the serum of CRC patients compared to healthy controls (30.41 ± 11.14 ng/mL, *p* < 0.0001) [105]. B7-H3 upregulation has been associated with several clinicopathologic characteristics, including disease stage (*p* < 0.001 [102], *p* = 0.032 [103], *p* = 0.025 [104]), histological differentiation (*p* = 0.017 [106], *p* = 0.0227 [105]), and lymph node infiltration (*p* = 0.023 [103], *p* = 0.034 [107]). These associations can be explained by some main functional effects of B7-H3 described in preclinical studies. In mice, the administration of CRC cell-derived B7-H3 rich exosomes promoted metastasis of CRC cells by activating the AKT1/mTOR signaling pathway [108]. B7-H3 upregulation was associated with increased MMP-9 levels, through stimulation of the JAK2/STAT3 pathway, resulting in pro-migratory and pro-invasive abilities [65]. In vitro, miR-29a upregulation reduced B7-H3 expression in CRC cells and reduced their invasive and migratory abilities [103]. Upon downregulation of B7-H3, a marked reduction in the proliferation of CRC cell lines along with a decrease in important key cell cycle-related proteins such as cyclin D1 and CDK4 were observed [47]. B7-H3 is also implicated in the regulation of CRC metabolism as it is significantly correlated with IDH1 levels and promotes aerobic glycolysis through hexokinase 2 induction both in vivo and in vitro [67]. Furthermore, B7-H3 can promote CRC resistance to chemotherapy, including oxaliplatin and 5-fluorouracil via regulation of CDC25A expression through the STAT3 signaling [64]. B7-H3 inhibited doxorubicin-induced cellular senescence of CRC cells in vivo. High expression of B7-H3 prevented doxorubicin-induced cellular senescence and growth arrest of CRC cells in vivo through the AKT/TM4SF1/SIRT1 pathway, while the knockdown of B7-H3 had the opposite effect [63]. CRC radioresistance has also been associated with B7-H3 expression, an effect attributable to B7-H3 regulation of KIF15-activated ERK1/2 [59]. Regarding prognosis, Gao et al. evaluated two European cohorts of 1244 patients in total and one Asian cohort of 179 patients and concluded high B7-H3 expression was strongly associated with OS even after adjusting for the effects of other expressed checkpoint molecules [109]. Concurrent low PD-L1 and B7-H3 expression has been associated with better OS when compared with high expression of either of these ICs (43.3 vs. 24.6 months, *p* < 0.01) [110].

Zekri et al. developed an IgG-based B7-H3xCD3 bispecific antibody and reported potent anti-tumor activity after both in vitro and in vivo testing [111]. TAA06 is a CAR T-cell injection targeting B7-H3 and its efficacy and safety are tested in a phase I trial, that is currently recruiting, in patients with CRC and other advanced solid malignancies (NCT05190185). A different phase I trial uses engineered TILs/CAR-TILs targeting various antigens with B7-H3 among them to treat CRC and other solid tumors (NCT04842812).

## 3. Gynecological Malignancies

### 3.1. Cervical Cancer (CC)

B7-H3 has been shown to be constitutively expressed in several gynecological malignancies including cervical [112,113], endometrial [114,115], and ovarian carcinoma [116,117]. When compared with normal cervical tissue, Li et al. found significantly higher immunostaining of B7-H3 in cancerous samples (72.22% vs. 15.00%, *p* < 0.001) [118]. More specifically, B7-H3 was detected in 94.0% of stromal compartments and up to 62.8% of tumor cells [30], with higher frequency in squamous cell carcinoma compared to adenocarcinoma (*p* = 0.001) [30,119]. In cases with mixed squamous carcinoma and adenocarcinoma, B7-H3 positivity was significantly associated with tumor size (*p* = 0.013) [120], deep stromal invasion (*p* = 0.0013) [53], and lymph nodal involvement (*p* = 0.032) [119]. Interestingly, in the study of Zong et al. where only adenocarcinomas were included, B7-H3 was not associated with any other clinicopathological features [112]. In squamous histology, these associations can be partially explained by the pro-proliferative and anti-apoptotic effects of B7-H3, as evidenced by the downregulation of tumor-suppressor Rb protein and the upregulation of the HPV oncogene E7, as well as the regulation of the apoptosis-related proteins P21, P16, PARP-1, Caspase-8, Bax, Bcl-2, and Bcl-xl [53]. In CC TME, B7-H3 holds a pleiotropic immunosuppressive role as it is inversely correlated with the number of CD8^+^ TILs (*p* < 0.0001) [113] and the expression of the proinflammatory cytokine IL-2 [120]. B7-H3 further restricts the antitumor immune response via enhanced IL-10 and TGF-B1 production [53]. A stimulatory effect of B7-H3 on FOXP3 (*p* < 0.001) has also been reported, adding to its immunosuppressive role [120]. Furthermore, B7-H3 was shown to be positively correlated with the number of cancer stem cells in CC cell lines, while it also conferred resistance to cisplatin [121]. Overall, these effects contribute to the dismal prognosis of B7-H3^+^ cervical malignancies as B7-H3 positivity is associated with shorter RFS (*p* = 0.006), OS (*p* = 0.023) [119], and CC-specific survival [30], independently of FIGO stage. In the study of Yang et al., miRNA-199a restricted the expression of B7-H3, via binding to the 3′-UTR of B7-H3 gene, thus inhibiting proliferation, migration, and invasion of CC cells. These findings were subsequently replicated in vivo in mouse xenograft models [122]. Despite this preclinical background, anti-B7-H3 modalities have not been entered yet in clinical phasing.

### 3.2. Endometrial Cancer (EC)

In EC, B7-H3 expression was reported in 94% [114] and 77.7% [115] of serous histology, but only in 18.5% of endometrioid cases [115]. Specimens with endometrial hyperplasia had no B7-H3 expression [115]. Importantly, B7-H3 expression in serous EC was significantly higher than TIM-3 and PD-L1 [114]. However, few data exist on the functional implications of B7-H3 overexpression. Brunner et al. pinpointed increased B7-H3 expression was associated with limited percentages of TILs (*p* = 0.017) [115]. Furthermore, B7-H3 positivity emerged as a high-risk feature, linked with higher likelihood of nodal metastasis (*p* = 0.05) [123] and significantly lower 5-year OS% (56% vs. 85%, *p* = 0.005) [115]. Ifanatamab deruxtecan (DS-7300) is another anti-B7-H3 ADC that is comprised by an anti-B7-H3 IgG1 mAb tethered to an exatecan (topoisomerase I inhibitor) derivative and is currently investigated for the treatment of advanced solid tumors, including EC, in a phase I/II clinical trial (NCT04145622). In preclinical testing, DS-7300 exhibited potent anti-proliferative activity against a human EC cell line (MFE-280). Importantly, DS-7300 demonstrated potent antineoplastic activity against an MFE-280 xenograft model in lower doses compared to other B7-H3 models [124]. In preliminary results from the NCT04145622 trial, DS-7300 exhibited a manageable toxicity profile with grade 3 or 4 TRAEs occurring in 20% of study population. Partial responses (PR) were observed in 13 of the 56 patients, including one EC case [125].

### 3.3. Ovarian Cancer (OC)

B7-H3 expression has been consistently documented in OC by a plethora of studies [116,117,126,127,128]. In the study of Zhang et al., B7-H3 was present in 93% of OC samples, while B7-H3 was not present in any of the 14 non-neoplastic ovarian specimens [126]. B7-H3 was found primarily on stromal cells, while tumor cell expressed B7-H3 to a lesser degree [117]. Higher B7-H3 expression was observed in the tumor vasculature of serous OC compared to other histological subtypes [126]. In OC, B7-H3 has been shown to hold a multi-functional immunosuppressive role. In high-grade serous OC, B7-H3 expression was higher in the proliferative and mesenchymal subtypes compared to the immunoreactive subtype (*p* = 0.004 and *p* < 0.001, respectively) [128]. In the same study, B7-H3 expression was correlated with M2 polarization of macrophages (*p* = 0.018) and decreased T-cell-produced IFN-γ (*p* = 0.047). B7-H3 expression was also positively correlated with Treg infiltration [116] and negatively correlated with NK-cell cytotoxicity in vitro and in vivo [129]. In the latter study, B7-H3 was also associated with OC cell glycolysis supporting its role in tumor metabolism. Furthermore, B7-H3 has been shown to promote therapeutic resistance, potentially through induction of the pro-survival PI3K/AKT/BCL-2 signaling pathway [69]. B7-H3 inhibition effectively reversed insensitivity to PD-L1 or paclitaxel [130].

## 4. Breast Cancer (BC)

B7-H3 has been observed to be aberrantly expressed in different subtypes of breast cancer in numerous studies. Cong et al. reported a significantly higher B7-H3 expression rate in BC specimens of various histological subtypes compared to adjacent healthy tissues (56.8% vs. 43.2%, *p* < 0.05) [131]. In DCIS, B7-H3 was expressed in 73.4% of cases and was found to be negatively correlated with CD3^+^and CD8^+^TILs’ density and positively associated with high-nuclear grade and comedo-type necrosis (*p* < 0.05) [132]. Higher expression rate was reported by Liu et al. who identified B7-H3 presence in 106 BC specimens from a total of 117 specimens (90.6%), consisted of 97 invasive ductal and 20 invasive lobular carcinomas. In the same study, B7-H3 expression correlated with lymph node infiltration (*p* = 0.018) and advanced disease (*p* = 0.011) [133]. In phyllodes tumor samples, B7-H3 levels correlated with disease progression, gradually increasing during malignant transformation of initially benign or borderline cases (r = 0.411, *p* < 0.001 and r = 0.293, *p* = 0.003, respectively) [134].

Regarding its functional implications, B7-H3 has been shown to contribute to immune evasion, disease progression, and treatment resistance. B7-H3^+^ lesions demonstrated markedly lower CD3^+^ and CD8^+^TILs (*p* = 0.001 and *p* = 0.027, respectively) [134]. This inverse relationship was further demonstrated in the study of Kim et al. [135] and Cheng et al. [136] in triple-negative disease as well as in an analysis of baseline characteristics of HER+ patients included in the TRYPHAENA clinical trial (NCT00976989) (*p* < 0.001) [137]. Importantly, Cheng et al. showed B7-H3 overexpression in tumor-associated macrophages is strongly associated with the formation of an immunosuppressive TME, enriched in Tregs and MDSCs and depleted of CD8^+^T-cells and NK cells [136]. In the presence of B7-H3 expression on BC cells, TILs experience restricted proliferation (*p* < 0.001) and IFN-γ production (*p* < 0.001), attributed to decreased PIK3/AKT/mTOR signaling [39]. mTOR comprises a major determinant of CD8^+^T-cell functional fate, driving their differentiation into effector rather than memory cells [138]. Importantly, constitutive activation of the mTOR pathway is also pertinent to BC progression, as its activation promotes anabolism [139] and confers enhanced proliferation and metastatic potential [140]. PI3K/Akt/mTOR inhibitors comprise a validated treatment modality, especially for hormone receptor positive/HER2 negative disease [141]. B7-H3 overexpressing BC cells were found to be less susceptible to the effects of the PI3K/Akt/mTOR pathway inhibitors, triciribidine and everolimus, compared to B7-H3 knock-out counterparts [142]. B7-H3-induced chemoresistance was also described upon paclitaxel administration, as B7-H3 upregulation largely restricted the cytotoxic effect of paclitaxel through activation of the anti-apoptotic JAK2/STAT3 pathway [61]. B7-H3 silencing resulted in significantly decreased anti-apoptotic JAK2/STAT3 downstream and enhanced sensitivity to paclitaxel. B7-H3-incited BC stem cell expansion, through the MAPK/ERK signaling axis, constitutes another possible mechanism of uprising resistance [46]. Activation of the MEK/ERK pathway was shown to promote lung metastasis of BC in vivo. [143]. Many studies have pinpointed B7-H3 as a negative prognostic biomarker associated with metastatic disease and worse outcomes. B7-H3 positivity was significantly associated with lymph node infiltration (*p* = 0.004), while B7-H3 mRNA emerged as a predictor of lymph node metastasis (*p* = 0.021) [144]. High expression of B7-H3 was associated with worse OS in the studies of Fang et al. (*p* = 0.048) [145] and Kim et al. (*p* < 0.01) [135]. Patients with B7-H3 high lesions also experienced a significantly lower 5-year RFS% (76.3% vs. 94.7%, *p* = 0.0137) [146] and 5-year DFS% (86.7% vs. 92.4%, *p* = 0.027) [147]. B7-H3 was also validated as an independent predictor of RFS through a multivariate analysis (*p* = 0.025) [146].

The more aggressive BC behavior in the presence of B7-H3 has propelled anti-B7-H3 modalities into preclinical and recently clinical testing. As previously discussed, B7-H3 inhibition enhances the anti-tumor effects of PI3K/AKT/mTOR [61] and increases sensitivity to paclitaxel [61]. Cheng et al. treated BC bearing mice with anti-B7-H3 in combination with paclitaxel or anti-PD-1 mAb and concluded B7-H3 inhibition significantly potentiated the tumoricidal effects of either paclitaxel or anti-PD-1 as monotherapies [136]. In a clinical trial setting, enoblituzumab is currently tested for patients with TNBC in two phase I trials either as a monotherapy (NCT01391143) or in combination with ipilimumab (NCT02381314). CAR-T cells targeting B7-H3 for breast cancer are also evaluated in two phase I trials (NCT04842812, NCT05341492). Interestingly, microbubbles functionalized with B7-H3-targeted affibody were developed by researchers and exhibited enhanced B7-H3 molecular signal in breast tumors, contrary to normal breast tissue, underscoring the clinical value of B7-H3 in BC diagnosis [148].

## 5. Urologic Malignancies

### 5.1. Renal Cell Carcinoma (RCC)

In RCC, B7-H3 has been mainly proposed as a disease-specific endothelial marker with a reported expression of 95% [149] and 98% [150] in tumor vasculature. In the same studies, tumor cells were B7-H3^+^ in 17% and 19% of specimens, respectively. B7-H3 expression has also been observed in further RCC stromal compartments [151,152]. A higher positivity rate was reported by Nunes-Xavier et al., as B7-H3 was expressed in 40% of tumor cells from RCC specimens of various histological subtypes [153]. Importantly, in Wilms tumor, B7-H3 expression was identified in 100% of stained specimens, of which 67% were highly positive, pinpointing B7-H3 as a therapeutic target in pediatric malignancies beyond neuroblastoma [154].

B7-H3 expression confers multifaceted implications in RCC natural history and prognosis. In the TME, B7-H3 expression on cancer-associated fibroblasts promotes their proliferation and augments secretion of pro-tumorigenic proteins including HGF and CXCL-12, ultimately driving disease progression [152]. The notable B7-H3 presence in RCC endothelial cells presumably induces angiogenesis, through interacting with CD14^+^monocytes [155] and mediating a Tie-2-dependent activation of the NF-kb/VEGF pathway [156]. Diffuse vascular B7-H3 expression correlated with advanced TNM stage (*p* < 0.001) [150] and increased risk of death from RCC (risk ratio: 1.38, *p* = 0.029) [149]. B7-H3 was further correlated with synchronous metastasis (*p* = 0.007) and poorer PFS (*p* = 0.031) [157]. B7-H3 is also implicated in immune evasion as its expression, on either tumor cells or vasculature, is significantly associated with the number of FOXP3^+^Tregs (*p* = 0.041 and *p* = 0.0007, respectively) and a higher disease-specific mortality (*p* = 0.0084 and *p* = 0.017, respectively) [45]. High numbers of B7-H3^+^ and FOXP3^+^TILs emerged as independent negative prognostic factors for local recurrence and metastasis post-radical nephrectomy (*p* = 0.006 and *p* = 0.033, respectively) [158].

Based on these data, several anti-B7-H3 modalities have been tested in preclinical and lately clinical trial setting. It should be noted the higher presence of B7-H3 in RCC vasculature compared to tumor cells rendered it a suitable target for drugs targeting the tumor stroma and especially for the promising and highly efficacious stroma-targeting, internalizing, or non-internalizing ADCs [31]. To this end, Seaman et al. presented an ADC consisting of an anti-B7-3 mAb tethered to a pyrrolobenzodiazepine payload that demonstrated significant dual-compartment ablation of both tumor cells and vasculature eradicating both primary solid tumors and metastatic sites [151]. Vobramitamab duocarmazine is currently being tested in combination with lorigerlimab in a phase I trial (NCT05293496) in patients with solid malignancies, including RCC. Furthermore, Wang et al. described an anti-B7-H3 mAb labeled with ^131^I (^131^I-4H7) that produced significant tumoricidal effects in mouse models of RCC [159]. Bivalent CAR T-cells targeting CD70 and B7-H3 were found to be more effective than unspecific CAR T-cells in restraining tumor growth in RCC cell lines [160]. In Wilms tumors, phase I trials evaluating CAR T-cell treatment (NCT04483778, NCT04897321) and Enoblituzumab (NCT02982941) are ongoing.

### 5.2. Prostate Cancer (PC)

The B7-H3 gene was documented in the top 19th percentile of PC expressed genes [161], with the respective coding B7-H3 checkpoint molecule being significantly upregulated compared to further relevant ICs including PD-L1, PD-L2, and B7-H4 [161,162]. In a large analysis of more than 12,000 PC specimens, B7-H3 positivity was documented in 47% of tumor samples, while normal prostatic glands showed minimal B7-H3 expression [163]. B7-H3 expression was also significantly higher in cancerous tissue compared to benign prostatic hyperplasia samples (174.73 ± 56.80 vs. 82.69 ± 46.19 ng/g, *p* < 0.001) [164]. Intense B7-H3 staining was associated with advanced stage (*p* < 0.0001), high Gleason score (*p* < 0.0001), lymph node infiltration (*p* < 0.0001), and positive surgical margins (*p* = 0.0015) [163]. Increased B7-H3 was consistently associated with advanced disease in further studies [161,162,165] and with upregulated expression of androgen receptor (*p* = 0.016 [162], *p* < 0.0001 [163]). In the TME of PC, B7-H3 prevents the apoptosis of MDSCs, and restricts the infiltration of CD8^+^T-cells, and NK cells, leading to the formation of an immunosuppressive tumor milieu. Patients with B7-H3^+^ disease were significantly more likely to have metastatic disease compared to B7-H3^−^ subjects (31% vs. 12%; *p* = 0.0003), while they also had a higher disease-specific mortality rate.

B7-H3-targeting modalities have emerged as a promising therapeutic option with exciting early clinical data that have upgraded anti-B7-H3 agents into late clinical testing. Vobramitamab duocarmazine is currently utilized in three clinical trials in patients with metastatic castrate resistant prostate cancer (mCRPC) alone or in combination with other agents (NCT05293496, NCT05551117, NCT03729596). Preliminary data from the phase I trial NCT03729596 that evaluates Vobramitamab duocarmazine with or without the anti-PD-1 mAb retifanlimab in patients with mCRPC yield a good safety profile, a ≥50% PSA reduction in eleven of twenty-two evaluable patients and an unconfirmed PR in four out of seven evaluable patients [166]. Furthermore, administration of another anti-B7-H3 ADC, DS-7300, was well-tolerated and achieved a PR in 10 out of 29 enrolled subjects. Neoadjuvant enoblituzumab was used in patients with localized PC prior to prostatectomy in a phase II trial (NCT02923180) with no serious TRAEs and an undetectable PSA level (<0.1 ng/mL) one year after prostatectomy in 66% [167].

## 6. Lung Cancer

B7-H3 has been observed to be expressed in both small cell lung cancer (SCLC) and non-small cell lung cancer (NSCLC). In SCLC, B7-H3 was expressed in 64.49% [168] and 64.9% [169] of patient specimens, while in NSCLC, the reported expression rate ranged between 69.5% and 80.4% [170,171,172]. Among NSCLC subtypes, B7-H3 expression was significantly more common in squamous carcinoma compared to adenocarcinoma (95% vs. 49%, *p* < 0.001 [173], *p* = 0.048 [170], *p* = 0.004 [174]). B7-H3 was expressed in 88.3% of resected squamous lung carcinoma [175]. Contradictory data exist on the association of B7-H3 with disease stage both in SCLC and NSCLC [169,172,174,176].

B7-H3 expression was associated with the intratumoral density of Tregs, shaping an immunosuppressive TME. B7-H3 high/FOXP3 high patients had significantly worse OS compared to B7-H3 low/FOXP3 low counterparts (*p* = 0.006) [174]. Furthermore, B7-H3 was significantly upregulated in tumor-residing DCs, compared to healthy lung tissue DCs, and markedly restricted their T-cell costimulatory activity [40]. B7-H3 also promotes lung cancer EMT as suggested by its correlation with the mesenchymal marker n-cadherin and vimentin [50]. Vasculogenic mimicry, which is an alternative microvascular circulation independent of angiogenesis, is promoted B7-H3 expressing NSCLC tumor cell that act via PI3K/AKT signaling pathway [177]. Furthermore, B7-H3 orchestrates SCLC metabolic reprogramming as it enhances glucose uptake, extracellular acidification rate, and oxygen consumption rate, while its deletion significantly decreases the activation of several oncogenic signaling pathways, including ERK, AKT, and STAT3 [178]. B7-H3 also promotes lipid metabolism through upregulation of fatty acid synthases in lung cancer [68]. In lung adenocarcinoma patients treated with EGFR TKIs, B7-H3 emerged as independent predictor of response through both univariate and multivariate analysis. In the B7-H3 high group ORR was 16% compared to 74.2% in the B7-H3 low group (*p* < 0.001). Similarly, the mOS was significantly worse in B7-H3 high patients (15.9 vs. 25.7 months, *p* = 0.03) [176]. Poor survival upon PD-1 inhibition in B7-H3^+^ NSCLC patients was reported by Yim et al. (*p* = 0.026) [179]. The association of B7-H3 expression with OS in NSCLC has been consistently reported in several other studies [171,172]. In SCLC, B7-H3^+^ patients have a significantly worse OS compared to the negative cohort (7.39 vs. 23.81 months, *p* = 0.019) [168].

In case of lung cancer, anti-B7-H3 modalities have also produced some encouraging preliminary clinical results. Enoblituzumab is an anti-B7-H3 mAb currently evaluated as monotherapy or in combinatorial regimens in three clinical trials in patients with NSCLC and other solid tumors (NCT02381314, NCT01391143, NCT02475213). Dual B7-H3 and PD-1 inhibition with enoblituzumab and pembrolizumab showed safe profile along with promising efficacy in NSCLC and other solid tumor patients (NCT02475213). Out of the 14 evaluable anti-PD-1/PD-L1 naïve NSCLC patients an ORR was observed in 35.7% (5/14), all of which were PRs, with a mean DoR of 8.3 months. In an additional eight patients SD was observed (57.1%). Of the twenty-one patients with NSCLC previously treated with anti-PD-1/PD-L1 agents, eleven (52.4%) had SD and two (9.5%) achieved a PR [180]. A phase I trial (NCT02381314) evaluates the safety of enoblituzumab in combination with ipilimumab in patients with NSCLC and other B7-H3 expressing solid malignancies (NCT02381314). HS-20093, is a novel B7-H3-targeting ADC, that is currently tested as monotherapy in patients with various solid malignancies in another phase I study (NCT05276609). Results from this study, which included twenty-nine patients with NSCLC and eleven patients with SCLC, demonstrated an acceptable safety profile along with promising antitumor activity, especially in the SCLC cohort where PRs were noted in seven out of nine patients [181]. Ifinatamab Deruxtecan is now examined in patients with pretreated extensive-stage SCLC in an active phase II clinical trial (NCT05280470). Various B7-H3-targeting CAR T-cells have been developed up to phase I studies in patients with lung cancer and other solid malignancies (NCT05190185, NCT03198052, NCT05341492).

## 7. Neuroblastoma

The unmet need for more effective therapeutic options in neuroblastoma has led to testing of anti-B7-H3 agents in preclinical and clinical setting. Although robust B7-H3 expression in neuroblastoma specimens has been consistently reported in many studies [41,154,182], relatively few data have been published regarding its functional implications. Ni et al. demonstrated B7-H3 upregulation enhanced STAT3 and PI3K signaling and conferred heightened neuroblastoma proliferation, invasiveness, and migration [183]. In neuroblastoma mouse cell lines, B7-H3 repressed NK cytotoxic potential [41]. Increased B7-H3 expression was associated with decreased OS in 1864 cases of neuroblastoma (*p* < 0.05). In the same study, B7-H3 knockdown effectively inhibited neuroblastoma cell proliferation [182]. Grote et al. showed second-generation B7-H3-targeting CAR-engineered NK cells exerted specific and long-term elimination of neuroblastoma cells in vitro without affecting B7-H3-negative cancer cells [184]. B7-H3-CAR NK-92 cells were also found to display increased cytotoxicity in a three-dimensional neuroblastoma spheroid model. This can recapitulate in vivo morphology, cell connectivity, gene expression, polarity, and tissue architecture, effectively bridging the gap between in vitro and in vivo models, while a CAR-NK trial is about to start. In neuroblastoma mouse models, the anti-B7-H3 ADC, m276-SL-PBD, achieved disease response in 91.5% of cases [185]. Heubach et al. suggested Fc-optimized anti-B7-H3 mAbs and anti-B7-H3 ADCs could augment the therapeutic outcomes in GD2^−^ tumors [186]. Du et al. [89] and Birley et al. [187] demonstrated B7-H3 CAR T-cells anti-tumor potency in vitro and in xenograft mice neuroblastoma models. Recently, Tian et al., aiming to overcome the variability in expression of GPC2 and B7-H3 on neuroblastoma cells, constructed a bicistronic CAR, which compared to classic single antigen CAR T-cells showed enhanced anti-tumor efficiency, longer T-cell endurance and resilience to exhaustion both in vitro and in vivo [188]. Hernandez et al. successfully manufactured bispecific mAbs with high avidity to GD2^+^/B7-H3^+^ and limited off-target binding, suggesting therapeutic potential and a manageable toxicity profile [189].

In clinical testing, radioimmunotherapy using ^131^I-omburtamab yields a favorable safety profile in a phase I trial (NCT00089245) and is currently tested in further phase I (NCT05064306) and phase II/III (NCT03275402) studies. Enoblituzumab is tested as a therapeutic candidate in a phase I trial (NCT02982941). B7-H3 CAR T-cell treatment showed encouraging anti-tumor activity and manageable toxicity profile in pediatric relapsed/refractory solid tumors, including neuroblastoma in a phase I trial (NCT04483778) [190]. Additional CAR T-cell therapies are evaluated in ongoing phase I trials for the treatment of neuroblastoma (NCT04897321, NCT04637503, NCT04864821, NCT04691713, NCT04432649).

## 8. Melanoma

Following the high efficacy of first-generation CPIs, the expression of further checkpoint molecules beyond PD-1 and CTLA-4 and their involvement in the immune manipulation has become the focus of extensive preclinical and clinical research [5]. Although the marked immunogenicity of melanoma initially stimulates an effective antitumor immune response, the expression of IC molecules is a well-documented phenomenon that leads to T-cell exhaustion and allows for immunosurveillance evasion and immunotherapy resistance [4]. B7-H3 mRNA expression was found to be significantly higher in human melanoma tissues compared to melanocytic nevi and normal skin (*p* < 0.05), and was consistently positively associated with disease stage and metastatic dissemination (*p* < 0.05) [191]. Except for melanoma cells, B7-H3 was also expressed on the surface of melanoma-infiltrating macrophages and DCs and was shown to bind to infiltrating NK and CD8^+^T-cells [74]. In the last study, B7-H3 silencing significantly enhanced the proliferation and cytotoxic function of melanoma antigen-specific CD8^+^T-cells, as determined by an increase in IFN-γ and granzyme B secretion. Similarly, a substantial increase in NK cell absolute number and killing activity was reported upon B7-H3 KO [74]. Furthermore, B7-H3 expression has been implicated in promoting melanoma invasiveness and migratory capacity [191,192], as evidenced by decreased levels of metastasis-associated molecules, including MMP-2, STAT3, and IL-8. Metastatic melanoma cells with knockdown expression of B7-H3 showed modest decrease in proliferation and glycolytic capacity and were more sensitive to dacarbazine chemotherapy and small-molecule inhibitors (e.g., MEK and AKT/mTOR inhibitors) [193]. Similar effects were observed in melanoma cells in the presence of an inhibitory B7-H3 monoclonal antibody, while the opposite was seen in B7-H3-overexpressing cells. In melanoma bearing mice, B7-H3 KO subjects experienced significantly longer median survival compared to WT mice (147 vs. 65 days, *p* < 0.001) [193]. In another study in melanoma mouse models, combining B7-H3 and PD-1 inhibition conferred greater anti-tumor effects compared to single antibody treatment. [74]. Enhanced T-cell cytotoxicity was described upon administration of an anti-CD3xB7-H3 bispecific antibody in human melanoma cell lines in vitro and in vivo [194]. Zhang et al. documented the efficacy of B7-H3-targeted CAR-T cells in melanoma in vitro and in xenograft mouse models [6]. B7-H3-targeted CAR-T cells are currently under investigation in three clinical trials (NCT05190185, NCT04483778, NCT04897321). Preliminary results from the active STRIVE-02 trial (NCT04483778) in children and young adults with relapsed or refractory solid tumors, including melanoma, have shown promising antineoplastic activity without dose-limiting toxicities [190]. Interim data from a phase I/II study (NCT02475213) on the combination of enoblituzumab with pembrolizumab in subjects with advanced solid malignancies yielded poor efficacy in the melanoma sub-cohort (ORR = 7.7%) [180]. Enoblituzumab was also evaluated together with ipilimumab in a melanoma cohort with progression under previous checkpoint inhibition in a completed phase I trial with no published results (NCT02381314). Vobramitamab duocarmazine is also tested in melanoma patients in two clinical studies in combination with lorigerlimab (NCT05293496, NCT03729596). Data from the phase I study (NCT05293496) yielded an acceptable safety profile with evidence of clinical activity in subjects refractory to more than two prior lines of checkpoint inhibition [166].

## 9. Conclusions

The identification of ICs and the incorporation of IC-targeting modalities in metastatic or adjuvant setting have set a paradigm shift in cancer therapeutics. Although, immunotherapeutic agents have been a prominent addition to our anticancer arsenal, primary or acquired resistance comprised a major challenge in the quest for improving treatment outcomes. The detection of high-sensitivity biomarkers, the clarification of the complex intratumoral dynamics that drive immune escape, and the development of immunotherapeutic combinations to specific patient sub-cohorts are key insights for advancing cancer patient care. B7-H3 has emerged as a promising checkpoint molecule that exhibits higher and more robust expression compared to currently known checkpoint molecules. This is of particular importance as it can be exploited for developing bispecific antibodies that limit on-target off-tumor effects and enhance selectivity. Furthermore, it also increases the therapeutic potential of ADCs, which optimally require homogeneous antigen expression. To this end, fast-growing data have highlighted the immunosuppressive and pro-tumoral effects of B7-H3 and have produced a surge of interest in anti-B7-H3 agents. Early preclinical and clinical results of B7-H3 inhibition have demonstrated encouraging anti-cancer activity but better understanding of the dynamic interplay of B7-H3 with the other TME constituents would be essential for its optimal targeting; while combined inhibition of B7-H3 with other ICs could synergistically overcome resistance to currently administered CPIs. Anti-B7-H3 modalities including circumvention of sB7-H3 and generation of non-internalizing ADCs would permit targeting of expressed B7-H3 by stromal compartments; and overcoming B7-H3-mediated resistance could be another perspective for novel effective strategies. Future studies will consolidate the prognostic and predictive role of B7-H3 and guide a more personalized and biomarker-driven approach.

## Figures and Tables

**Figure 1 vaccines-12-00054-f001:**
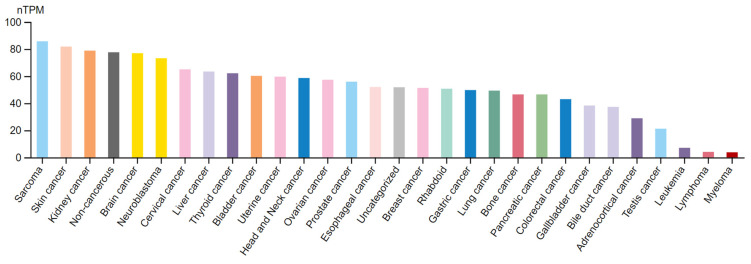
B7-H3 expression in cancerous tissue. Human B7-H3 mRNA expression in an array of malignancies in descending order of expression. nTPM: normalized transcript per million (Source: https://www.proteinatlas.org accessed on 15 December 2023).

**Figure 2 vaccines-12-00054-f002:**
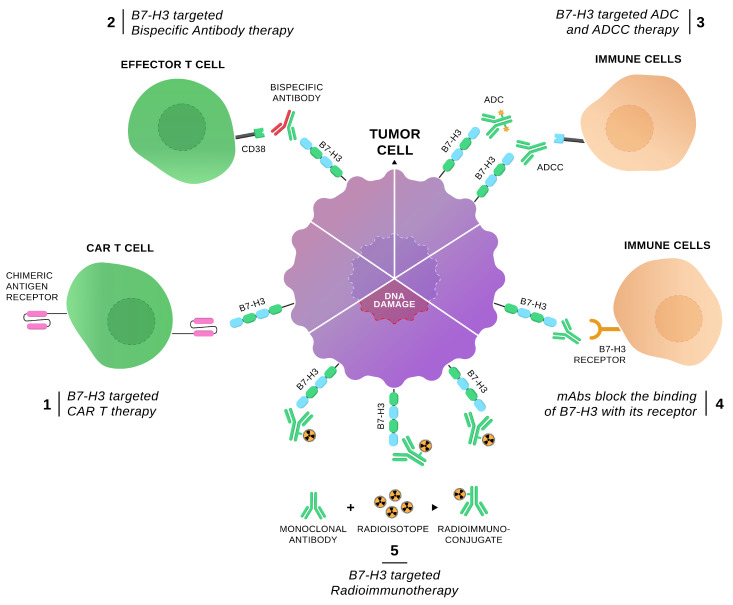
Summary of anti-B7-H3 modalities. Published approaches targeting B7-H3 include chimeric antigen receptor (CAR) T-cells, anti-B7-H3 bispecific antibodies, antibody-drug conjugates (ADCs), monoclonal antibodies, and antibody-dependent cell cytotoxicity as well as radioimmunotherapy.

**Figure 3 vaccines-12-00054-f003:**
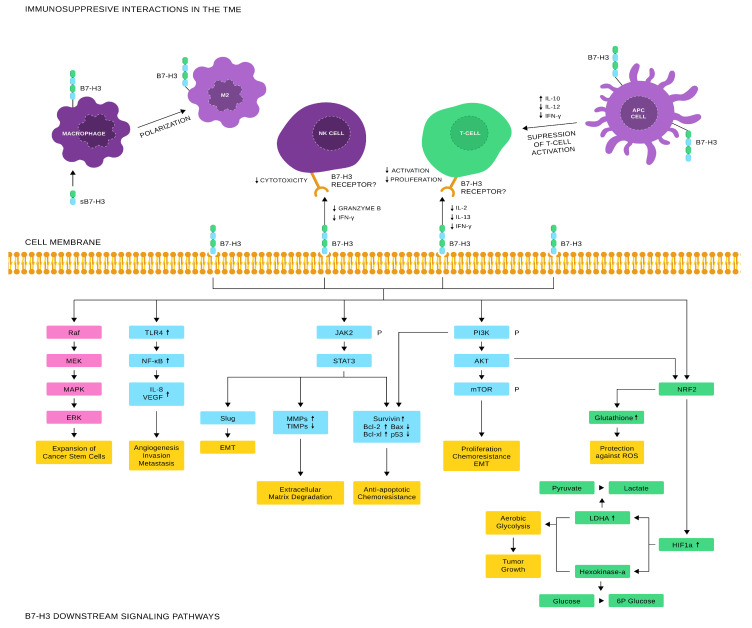
B7-H3 downstream signaling pathways and interactions with immune cells. B7-H3 induces M2 polarization of macrophages, restricts NK cell cytotoxicity, and T-cell activation and proliferation. In dendritic cells, B7-H3 reduces T-cell activation capacity. Within the cancer cells, JAK/STAT signaling results in epithelial-to-mesenchymal transition, chemoresistance, and apoptosis evasion. Stimulation of the TLR-4/NF-kB pathway promotes angiogenesis and heightens metastatic potential. The MAPK/ERK pathway expands the cancer stem cell population. Stimulation of the PI3K/AKT pathway confers multiple effects including, apoptosis evasion, chemoresistance, epithelial-to-mesencymal transition, protection against reactive oxygen species, and increased glycolysis. Abbreviations: EMT: epithelial-to-mesenchymal transition, ROS: reactive oxygen species; LDHA: lactate dehydrogenase.

## Data Availability

Data supporting the recommendations of this article are included within the reference list. Please contact the corresponding author for any further data request or Appendix A.

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
