# Peer review of "New Emerging Targets in Cancer Immunotherapy: The Role of B7-H3"

_vaccines, 2024, doi:10.3390/vaccines12010054_

Round 1
Reviewer 1 Report
Comments and Suggestions for Authors
In the review manuscript “New emerging targets in cancer immunotherapy: the roe of B7-H3”, Koumprentziotis et al. introduced the current insights about B7-H3 as a prognosis marker and therapeutic target. They collected comprehensive information over 13 diseases which is very useful. However, multiple roles of B7-H3 in the cancer treatment like target molecule, tumor cell modificatory, tumor microenvironment modificatory and immunosuppressive molecule are described a little bit disorganized. Re-organization of introduction part would make this manuscript more informative.
Major comments:
1. In introduction, various functions of B7-H3 are described (L78-L107, Table 1). These functions seem to be divided into B7-H3-expressing cell intrinsic ones and functions downstream of target molecules on the target cells. Please clarify which cell expresses B7-H3 and which cells expresses ligand/receptor molecules and which function is within which cell. It may be helpful to make sub-headings like tumor cell-intrinsic ones, TME-intrinsic ones and immune-suppression effect to immune cells like T cells.
2. In conclusion, what is the merits of B7-H3 as prognosis marker and as a therapeutic target compared to other immune checkpoint molecules? What are the major future tasks to develop B7-H3-targeted therapeutics?
Minor comments:
1. Figure 1 requires figure title and figure legend.
2. L206: Please add explanation for vobramitamab duocarmazine.
3. L207: Please add explanation for lorigerlimab.
4. L222: “resulting in apoptosis evasion of GC cells” should be inhibition of apoptosis and induction of evasion of GC cells.
5. L250: Please add explanation for trastuzumab.
6. L319: Please clarify that miRNA-199a does not bind B7-H3 molecule. It binds to 3’ UTR of B7-H3 gene (ref 92)
7. L385: constitutional activation may be miss-spelling of constitutive activation.
8. L458: unspecific may be miss-typing of unispecific.
Author Response
Reviewer 1:
General comments:
In the review manuscript “New emerging targets in cancer immunotherapy: the role of B7-H3”, Koumprentziotis et al. introduced the current insights about B7-H3 as a prognosis marker and therapeutic target. They collected comprehensive information over 13 diseases which is very useful. However, multiple roles of B7-H3 in the cancer treatment like target molecule, tumor cell modificatory, tumor microenvironment modificatory and immunosuppressive molecule are described a little bit disorganized. Re-organization of introduction part would make this manuscript more informative
.Authors’ reply: We thank the reviewer for his valuable feedback.
Major comments:
- In introduction, various functions of B7-H3 are described (L78-L107, Table 1). These functions seem to be divided into B7-H3-expressing cell intrinsic ones and functions downstream of target molecules on the target cells. Please clarify which cell expresses B7-H3 and which cells expresses ligand/receptor molecules and which function is within which cell. It may be helpful to make sub-headings like tumor cell-intrinsic ones, TME-intrinsic ones and immune-suppression effect to immune cells like T cells.
Authors’ reply: We have modified our introduction based on the reviewer’s suggestions and have added two subsections termed “Immunologic functions of B7-H3” and “Non-immunologic functions of B7-H3”, where we further elaborate on the data mentioned in table 1 in a more organized manner, as proposed. Preclinical studies evaluating the effects of B7-H3 on immune cells have used either B7-H3-Ig fusion proteins or B7-H3+ tumor cells, while providing indirect evidence for B7-H3 counterreceptors on the immune cells. An expanded presentation of proposed B7-H3 receptors has been added in the introduction, with regard to the cells expressing these molecules. Contradictory data exist regarding B7-H3/TLT-2 interaction, whereas little to no evidence has been published regarding specific functional implications of B7-H3/IL-20Rα and B7-H3/PLA2R1 interactions. Identification of B7-H3 receptors and elucidation of their downstream effects remains a very exciting research field that will aid to resolve the polarizing existing data on the roles of B7-H3.
- In conclusion, what is the merits of B7-H3 as prognosis marker and as a therapeutic target compared to other immune checkpoint molecules? What are the major future tasks to develop B7-H3-targeted therapeutics?
Authors’ reply: We have slightly expanded the conclusion in order to highlight the proposed points. "The identification of ICs and the incorporation of IC-targeting modalities in metastatic or adjuvant setting have set a paradigm shift in cancer therapeutics. Although, immunotherapeutic agents have been a prominent addition to our anticancer arsenal, primary or acquired resistance comprises a major challenge in the quest for improving treatment outcomes. The detection of high-sensitivity biomarkers, the clarification of the complex intratumoral dynamics that drive immune escape and the development of immunotherapeutic combinations to specific patient sub-cohorts are key insights for advancing cancer patient care. B7-H3 has emerged as a promising checkpoint molecule that exhibits higher and more robust expression compared to currently known checkpoint molecules. This is of particular importance as it can be exploited for developing bispecific antibodies that limit on-target off-tumor effects and enhance selectivity. Furthermore, it also increases the therapeutic potential of ADCs, which optimally require homogeneous antigen expression. To this end, fast-growing data has highlighted the immunosuppressive and pro-tumoral effects of B7-H3 and has produced a surge of interest in anti-B7-H3 agents. Early preclinical and clinical results of B7-H3 inhibition have demonstrated encouraging anti-cancer activity but better understanding of the dynamic interplay of B7-H3 with the other TME constituents is essential for its optimal targeting; while combined inhibition of B7-H3 with other ICs could synergistically overcome resistance to currently administered CPIs. Anti-B7-H3 modalities including circumvention of sB7-H3 and generation of non-internalizing ADCs will permit targeting of expressed B7-H3 by stromal compartments; and overcoming B7-H3-mediated resistance could be another perspective for novel effective strategies. Future studies will consolidate the prognostic and predictive role of B7-H3 and guide a more personalized and biomarker-driven approach.
Minor comments:
- Figure 1 requires figure title and figure legend.
Authors’ reply: We have added the missing title and legend of figure 1.
- L206: Please add explanation for vobramitamab duocarmazine.
Authors’ reply: Vobramitamab duocarmazine is an ADC currently under clinical trial testing for several B7-H3 positive malignancies. Since it is frequently mentioned in our manuscript, we had opted to discuss its constituents and provide a reference regarding its preclinical development in L169, which is the first time we refer to this drug.
- L207: Please add explanation for lorigerlimab.
Authors’ reply: We had already provided a short explanation of lorigerlimab in L171. We further added a reference from the work of Huang et al., which provides useful insights regarding DART molecules.
- L222: “resulting in apoptosis evasion of GC cells” should be inhibition of apoptosis and induction of evasion of GC cells.
Authors’ reply: We have integrated the suggested correction in our manuscript.
- L250: Please add explanation for trastuzumab.
Authors’ reply: We have added an explanation for trastuzumab, as suggested.
- L319: Please clarify that miRNA-199a does not bind B7-H3 molecule. It binds to 3’ UTR of B7-H3 gene (ref 92).
Authors’ reply: Thank you for pointing this out. We have rephrased the sentence accordingly.
- L385: constitutional activation may be miss-spelling of constitutive activation.
Authors’ reply: We have incorporated the suggested correction in our manuscript.
- L458: unspecific may be miss-typing of unispecific.
Authors’ reply: We have corrected the abovementioned misspelling.
Reviewer 2 Report
Comments and Suggestions for Authors
The review by Koumprentziotis et al. describes the function of B7-H3 in various tumours. The use of anti-B7-H3 therapeutics, alone or in combination with other common immunotherapeutics in solid tumours is clearly described.
Author Response
Reviewer 2:
General comments:
The review by Koumprentziotis et al. describes the function of B7-H3 in various tumours. The use of anti-B7-H3 therapeutics, alone or in combination with other common immunotherapeutics in solid tumours is clearly described.
Authors’ reply: We thank the reviewer for the positive general comments.
Reviewer 3 Report
Comments and Suggestions for Authors
In this review, the authors provide a comprehensive overview of the role of B7-H3 and its targeting in different types of cancer. After giving a brief overview of B7-H3 and its pleiotropic effects (tumor development and progression, anti-tumor immune response, treatment resistance, angiogenesis), the authors go into more detail on these effects and the therapeutic modalities currently under investigation in both preclinical and clinical settings in different cancer types. An effort has been made to summarize in a table (Table 1) the complex immunological and non-immunological functions of B7-H3, described in the literature according to cancer type.
Although very dense, the review is well-written, with a rich and recent bibliography.
The authors have chosen to present the functions of B7-H3 by type of cancer, which is useful for focusing on a particular pathology, but sometimes leads to a feeling of redundancy.
The review perhaps lacks illustration to facilitate reading, such as a figure describing B7-H3 and summarizing the various signaling pathways afferent to B7-H3 (PI3K/AKT/mTOR, JAK2/STAT3...)
In the introduction, presentation of B7-H3 could be improved : for instance the authors do not mention that B7-H3 may have different cellular localization profiles according to the cancer types, (membrane, cytoplasme, nucleus), or that an aberrant overexpression is observed not only on tumor cells but also on immune cells and stromal cells depending on the cancer type.
As well, the authors quickly mention the impact of soluble B7-H3. Is soluble B7-H3 observed in normal serum ? could the presence of circulating or intra-tumoral soluble B7-H3 be a problem for anti-B7-H3 therapies?
Do the anti-B7-H3 antibodies used in the different therapeutic strategies are blocking antibodies?
Legend of figure 1 is missing
Author Response
Reviewer 3:
General comments:
In this review, the authors provide a comprehensive overview of the role of B7-H3 and its targeting in different types of cancer. After giving a brief overview of B7-H3 and its pleiotropic effects (tumor development and progression, anti-tumor immune response, treatment resistance, angiogenesis), the authors go into more detail on these effects and the therapeutic modalities currently under investigation in both preclinical and clinical settings in different cancer types. An effort has been made to summarize in a table (Table 1) the complex immunological and non-immunological functions of B7-H3, described in the literature according to cancer type. Although very dense, the review is well-written, with a rich and recent bibliography. The authors have chosen to present the functions of B7-H3 by type of cancer, which is useful for focusing on a particular pathology, but sometimes leads to a feeling of redundancy.
Authors’ reply: We thank the reviewer for the positive general comments.
Specific comments:
- The review perhaps lacks illustration to facilitate reading, such as a figure describing B7-H3 and summarizing the various signaling pathways afferent to B7-H3 (PI3K/AKT/mTOR, JAK2/STAT3...)
Authors’ reply: According to the reviewer’s suggestion we have added another figure (Figure 2.) that illustrates downstream signaling upon B7-H3 engagement.
- In the introduction, presentation of B7-H3 could be improved: for instance, the authors do not mention that B7-H3 may have different cellular localization profiles according to the cancer types, (membrane, cytoplasm, nucleus), or that an aberrant overexpression is observed not only on tumor cells but also on immune cells and stromal cells depending on the cancer type.
Authors’ reply: A short discussion of B7-H3 localization had already been provided in some cases, such as in gastric cancer where the notable B7-H3 expression in the stromal compartment along with stroma-targeting ADCs as a therapeutic strategy are presented. However, taking into consideration the reviewer’s suggestion, we have modified the introduction of our manuscript and have added a subsection that briefly discusses the B7-H3 cellular localization and B7-H3 expression in normal and diseased cells under the respective subheading. Please note that we have enriched our bibliography accordingly.
- As well, the authors quickly mention the impact of soluble B7-H3. Is soluble B7-H3 observed in normal serum? could the presence of circulating or intra-tumoral soluble B7-H3 be a problem for anti-B7-H3 therapies?
Authors’ reply: We have integrated a short discussion regarding soluble B7-H3 in the subsection termed “B7-H3 localization”, with regard to the presence of soluble B7-H3 in normal serum and a potential impact on the effect of anti-B7-H3 agents. Please note that we have not included detailed evidence regarding prognostic significance as these data are elaborated in the respective sections.
- Do the anti-B7-H3 antibodies used in the different therapeutic strategies are blocking antibodies?
Authors’ reply: As depicted in figure 1, discussed anti-B7-H3 antibodies are blocking antibodies that prevent B7-H3 interaction with its putative receptors.
- Legend of figure 1 is missing
Authors’ reply: We have added the title and legend of Figure 1.
Reviewer 4 Report
Comments and Suggestions for Authors
Dr. Koumprentziotis et al. in this review article summarizes the potential role and therapeutic as well as translational advancement of B7-H3/ CD276 protein in the field of immunotherapy.
The review is very well written and reported. I have few comments which need to be addressed-
1- It will be easier for readers to understand the role of B7-H3 in cancer if there is a figure summarizing its expression and interaction with specific receptor and related function. Page-2, 65-70.
2- In figure-1 the writings associated with receptor ligand interactions are very small to read, please increase the front size.
3- It will be well organized and easy to go with the flow if there will be an introductory section in section-2 (page-6, 126).
Like-
2- Clinical significance and translational potential of B7-H3 in different cancer types-
Please add some data from protein atlas or TCGA about adverse survival benefit with higher expression of B7-H3.
RNA expression of B7-H3 in different types of cancer in one figure.
4- In neuroblastoma section please add and give reference related with CAR-NK trial about to start.
Author Response
Reviewer 4:
General comments:
Dr. Koumprentziotis et al. in this review article summarizes the potential role and therapeutic as well as translational advancement of B7-H3/ CD276 protein in the field of immunotherapy. The review is very well written and reported.
Authors’ reply: We thank the reviewer for the positive general comments.
Specific comments:
- It will be easier for readers to understand the role of B7-H3 in cancer if there is a figure summarizing its expression and interaction with specific receptor and related function. Page-2, 65-70.
Authors’ reply: Taking into consideration that B7-H3 receptors are still to be conclusively documented along with the contradictory data on the B7-H3/TLT-2 interaction we have opted to omit creating a figure that would illustrate these interactions as both their presence and their functional implications remain to be elucidated. However, we have added a supplementary table that summarizes the expression of B7-H3 in the discussed malignancies with regard to the specific intracellular localization and stromal presence. Furthermore, we have created another figure that illustrates downstream signaling pathways upon B7-H3 engagement.
- In figure-1 the writings associated with receptor ligand interactions are very small to read, please increase the front size.
Authors’ reply: Thank you for pointing this out. We have modified figure 1 accordingly.
- It will be well organized and easy to go with the flow if there will be an introductory section in section-2 (page-6, 126). Like clinical significance and translational potential of B7-H3 in different cancer types. Please add some data from protein atlas or TCGA about adverse survival benefit with higher expression of B7-H3. RNA expression of B7-H3 in different types of cancer in one figure.
Authors’ reply: We have added a figure from the protein atlas that displays rna expression of B7-H3. Furthermore, a table that demonstrates B7-H3 expression in the discussed malignancies has been created (probably as Supplementary Table 1). Evidence of reduced survival in cases of B7-H3 overexpression in solid malignancies is discussed in the respective subsections. Data from the TCGA have been included, when available and processed, as in HCC subsection, L242. Although an introductory part in section 2 would enhance coherence of our manuscript, this would inadvertently lead to repetition of data discussed in the following paragraphs and will possibly create a notion of redundancy. We have, however, expanded our introduction to provide a better presentation of B7-H3 functions, which will aid understanding of its role in each cancer.
- In neuroblastoma section please add and give reference related with CAR-NK trial about to start.
Authors’ reply: The preclinical evaluation and the discussion of CAR-NK cells in neuroblastoma has been modified as following: “Grote et al. showed that second-generation B7-H3-targeting CAR-engineered NK cells exerted specific and long-term elimination of neuroblastoma cells in vitro without affecting B7-H3-negative cancer cells [186].B7-H3-CAR NK-92 cells were also found to display increased cytotoxicity in a three-dimensional neuroblastoma spheroid model. This can recapitulate in vivo morphology, cell connectivity, gene expression, polarity, and tissue architecture, effectively bridging the gap between in vitro and in vivo models, while a CAR-NK trial is about to start.”
.
Round 2
Reviewer 1 Report
Comments and Suggestions for Authors
Thank you for responding appropriately to the comments. The manuscript has been significantly improved. The revised manuscript is now suitable for publication in Vaccine.
Author Response
We would like to thank the reviewer for his positive comments and his decision for our revised manuscript.